# Designing a User Study for Comparing 2D and VR Human-in-the-Loop Robot Planning Interfaces

Gregory LeMasurier
*University of Massachusetts Lowell*
gregory_lemasurier@student.uml.edu

Jordan Allspaw
*University of Massachusetts Lowell*
Jordan_Allspaw@uml.edu

Murphy Wonsick
*Northeastern University*
wonsick.m@northeastern.edu

James Tukpah
*Northeastern University*
tukpah.j@northeastern.edu

Taskin Padir
*Northeastern University*
t.padir@northeastern.edu

Holly A. Yanco
*University of Massachusetts Lowell*
holly@cs.uml.edu

Elizabeth Phillips
*George Mason University*
ephill3@gmu.edu

*Abstract*—**Human-in-the-loop robot teleoperation interfaces enable operators to control robots to complete complex tasks, as seen by the success of teams in the DARPA Robotics Challenge. In our prior work we discussed our virtual reality planning interface for performing dexterous robot teleoperation. In this work, we discuss plans and design for a user study to compare two human-in-the-loop planning interfaces, a 2D keyboard and mouse interface modeled after those used in the DARPA Robotics Challenge and a 3D virtual reality interface, for teleoperating a robot across navigation and manipulation tasks. In our study, we will compare operator performance, situation awareness, cognitive workload while using the interface, as well as the perceived usability of each. This work will contribute to building effective and intuitive teleoperation interfaces for controlling robots to complete complex tasks and in challenging environments.**

*Index Terms*—**Human-robot interaction (HRI), Robot Teleoperation, Virtual Reality (VR), Mobile Robots**

## I. Introduction and Related Work

The idea of using Virtual Reality (VR) to visualize or control robots has been present since early versions of VR Domes as early as 1993 [1]. However, it was not until the widespread availability of consumer grade VR headsets such as the Oculus Rift and the HTC Vive in 2016 that VR was poised to be more than just a theoretical interface for human-robot and other human-computer interactions. With commercial VR devices widely available, and the demand for remote robot operation higher than ever, researchers across the world are considering the merits of using VR for robot teleoperation [2]. Even though interest surrounding VR applications is rapidly growing, it is still necessary to demonstrate that VR is not just an exciting new technology, but indeed can provide many benefits for robot teleoperation across a variety of different tasks.

The DARPA Robotics Challenge (DRC) was a challenge whereby teams remotely operated humanoid bipedal robots to complete a series of disaster relief related tasks, such as opening and traversing through doors or turning a valve. During our analysis of the DRC [3] we found that, successful teams converged on a small set of strategies to do so. For example, it was common to have different interface configurations for each task. The amount of autonomy would also vary among the tasks: some strategies were largely teleoperated every step of the way, while others were more supervisory allowing operators to focus on high-level tasks while the robot proposed plans and executed some tasks autonomously. Among the interfaces analyzed was Team WPI-CMU's interface [4] which incorporated a human-in-the-loop planning based strategy. Their team's robot never fell or required restarts during the competition, demonstrating the capability of their system and strength of supervisory control interfaces.

Due to the nature of the DRC, it was not enough to simply be able to perform the tasks, it was also necessary to be able to do so quickly and reliably. This required efficient and capable interfaces to accomplish. Most teams used traditional devices to develop their interfaces for including keyboards, mice, and monitors. However, after the DRC, VR devices became widely commercially available leading to the desire to understand how a VR interface would compare against an exemplar interface used for the DRC.

Several groups have conducted studies on VR control of robots, however these studies typically have focused on manipulation tasks. Whitney et al. conducted a user study analyzing the task completion time, usability, and experienced workload when teleoperating a manipulator across four interface types: Direct Control, 2D Keyboard and Mouse, Positional Hand Tracking with Monitor, and a VR interface [5]. Ultimately, they found that their VR interface had a higher reported usability, lower workload, and operators were able to complete manipulation tasks faster than when using a keyboard and mouse interface [5]. Additionally, Hetrick et al. compared waypoint-like position controls to trajectory based controls in VR when remotely operating a manipulator [6]. They concluded that waypoint-like positional controls were more beneficial to enable novice operators to complete manipulation

tasks with the robot [6]. However, the interfaces developed in Hetrick et al. did not allow for any modifications to update the waypoint plans.

The contributions of this work include expanding upon the findings of VR and keyboard and mouse comparisons for direct control of robot systems, by comparing a VR and keyboard and mouse human-in-the-loop planning interface which communicates more information regarding the robot's plans to the operator. To our best knowledge a comparison of a VR and keyboard and mouse human-in-the-loop planning interfaces has not yet been conducted. Our interfaces also differ from the interfaces used in prior studies by incorporating aspects of supervisory control interfaces, where the operator, after confirming the plan, can observe the robot's actions and intervene when necessary. Additionally, we evaluate our interface on a mobile manipulator robot, thus allowing us to compare the interfaces on navigation tasks in addition to manipulation tasks. With this work, we hope to identify components from both interfaces that enable novice operators to effectively teleoperate and supervise robots to complete complex tasks in challenging environments.

In the following sections, we propose a user study to compare a 2D keyboard and mouse interface to a 3D VR interface to complete manipulation and navigation tasks for a mobile manipulator robot. We will measure task performance, perceived usability, and workload, to identify strengths and weaknesses of using VR interfaces versus traditional 2D keyboard and mouse interfaces, when remotely operating a mobile manipulator robot to perform complex manipulation and navigation tasks.

## II. SYSTEM DESIGNS

In our prior work, we introduced our virtual reality tele-operation interface [7]. This interface allows an operator to control a mobile manipulator robot from a remote location. Figure 1 displays the virtual robot's current state, the point cloud showing the environment in front of the robot, light blue functional waypoints for setting the state of the robot's end effector, and a visualization of the, represented by the turquoise robot, robot's plan through the waypoints.

The 2D interface, shown in Figure 2, was designed to be a simplified version of several of the interfaces developed for the DRC Finals and contains elements familiar on many user interfaces created for robotics researchers [8]. For example, teams such as Team ViGIR [9] and Team WPI-CMU [4] designed interfaces using a combination of buttons and sliders along with interactable markers. At the center of the screen is a large visualization window, showing a visualization of the robot, along with the point cloud, as seen in Figure 2. On both the right and left sides of the interface are a variety of controls to operate the robot.

Table I lists how information and planning actions are presented in both the VR and 2D interfaces. The interfaces are designed to each have the same functionality, but both utilize their respective mediums. Both interfaces use the same back end planners, and are capable of achieving the same actions to complete tasks (e.g., creating waypoints and confirming plans). For example, for the VR interface, we wanted to be able to take advantage of the ability to work in three dimensions, rather than forcing the user to only use 2D elements inside the virtual 3D world. For instance, controlling the robot's head in VR is done by reaching into the robot's head, pressing a button on the VR controller, and then dragging a target marker where you want the robot to look. This action is relatively fast, simple, and makes sense in the virtual reality task space. On the other hand to control the robot's head with a mouse and keyboard, the operator simply drags the robot's respective head sliders to command the robot to look up and down, or left and right. Forcing these actions to be done in a 3D interaction window could make the action unnecessarily complex. Thus, we allowed the interfaces to vary in ways which made intuitive sense given inherent strengths and weaknesses for visualizing information in 2D versus 3D. As such, each interface is not an exact replication of the other, but rather includes identical functionality for operating a robot to plan and execute manipulation and navigation tasks.

## III. HYPOTHESES

Through our proposed study, we plan to investigate the following hypotheses.

*Hypothesis 1: Usability* We hypothesize that participants will consider the 3D interface more usable than the 2D interface when controlling a robot to complete our experimental tasks. This follows findings from similar studies comparing traditional interfaces to VR interfaces [5].

*Hypothesis 2: Workload* We expect that our participants who complete tasks using the VR interface will report less workload than participants who complete tasks using the 2D interface due to the ability to interact in 3D using 3D devices versus 2D devices.

*Hypothesis 3: Situation awareness* We expect that participants who use the VR interface will develop higher levels of situation awareness than participants who use the 2D interface. Prior work by Pausch et al. found that users spent more time re-examining areas of rooms they had already searched using a 2D interface compared to a VR interface [10]. Because VR can allow users to easily acquire a complete visualization of the robot's operational space, we expect that participants who use the VR interface will develop higher levels of situation awareness than participants who use the 2D interface.

*Hypothesis 4: Task performance for manipulation and navigation tasks*

We will compare participants' rate of improvement relative to their own performance across task types in a given interface, rather than performance relative to others using a given interface. Therefore, we hypothesize the following: (a) participants' rate of improvement (i.e., the expected reduction in time it takes to complete tasks trial over trial) will be higher in the VR condition than in the 2D condition, (b) the number of collisions while completing the tasks to be fewer in VR, and (c) performance scores will be higher in the VR condition than in the 2D condition.

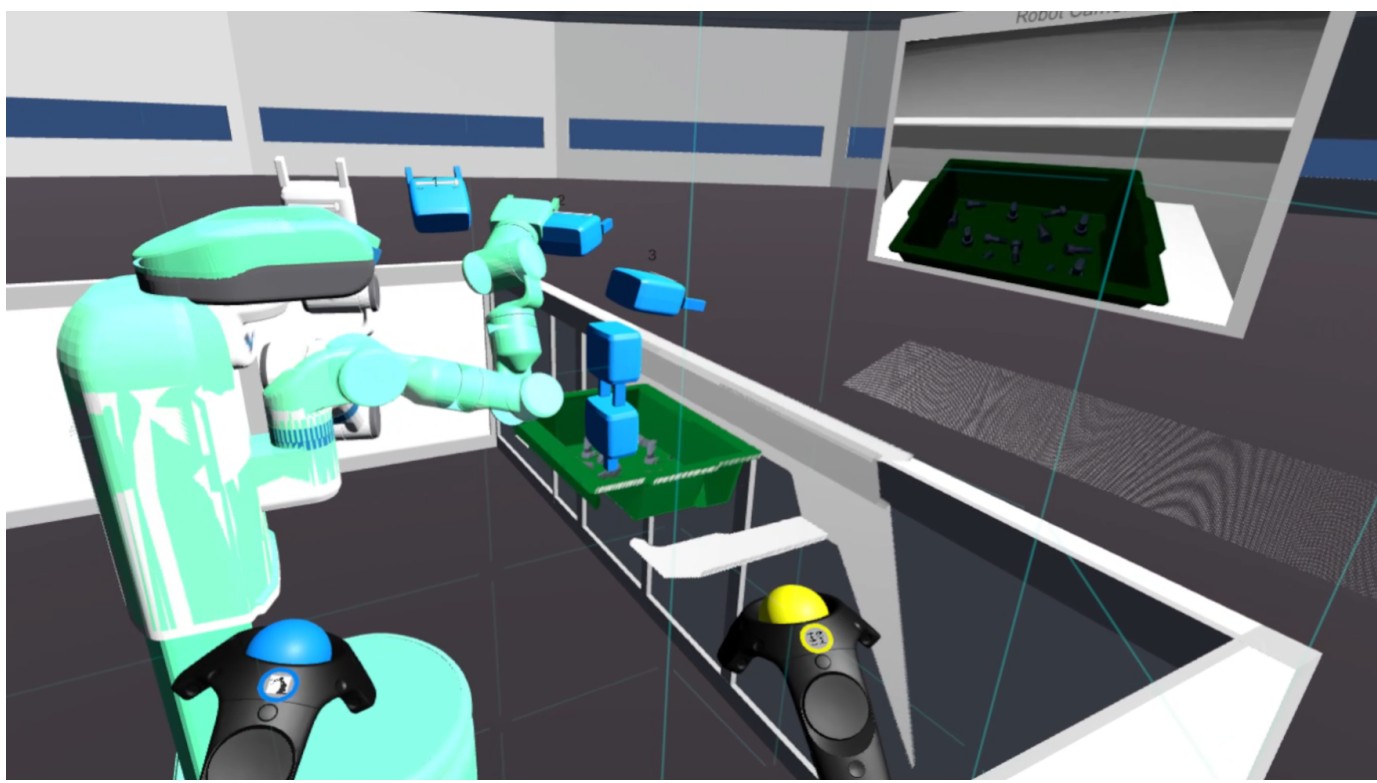

Fig. 1. The VR view of an operator watching the robot's planned trajectory, shown by the turquoise virtual robot, through the light blue waypoints set by the operator. In this figure the robot was commanded to pick an object out of the green bin, as seen in the point cloud and in the camera panel that is anchored to the front of the robot, as seen in the top right.

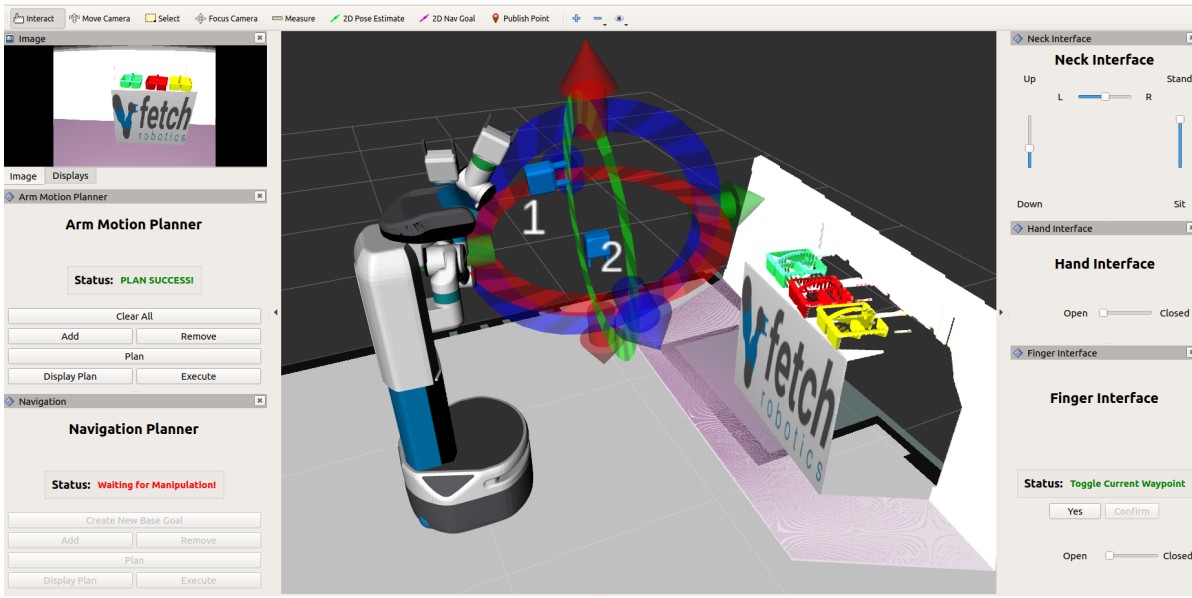

Fig. 2. Mouse and Keyboard interface. There are currently two manipulation waypoints, marked 1 and 2, where the currently selected waypoint is able to move on each rotation and cardinal axis. The navigation panel is disabled while the manipulation planner is active. The robot is currently demonstrating the path through the waypoints with the extra green arm.

| | Virtual Reality | Mouse and Keyboard |
|---|---|---|
| **Create Navigation Waypoint** | Operator clicks the "Create Navigation Waypoint" button in the wristwatch UI and a waypoint is placed in the scene. The operator can also press and hold the trackpad while pointing the controller, upon release a waypoint is placed. Finally, an operator can create a waypoint by holding an existing waypoint and pressing the trigger button to clone it. | Operator clicks the "Create Navigation Waypoint" located in the Navigation panel and a waypoint is placed in the scene. |
| **Adjust Navigation Waypoint** | Operator grabs a waypoint by reaching into a waypoint and pressing and holding the grip button on their controller. After grabbing the waypoint they can move it around to adjust its position and orientation. The operator can also drag the slider on a waypoint to raise or lower the target height at the waypoint. | Operator clicks a waypoint and uses marker controls (translational arrows or rotational scroll circle) to move the waypoint. |
| **Plan Navigation Path** | Operator clicks the "Plan Navigation Path" button in the wristwatch UI. | Operator clicks the "Plan Navigation Path" button in the Navigation panel. |
| **Execute Navigation Plan** | Operator clicks the "Execute Navigation Plan" button in the wristwatch UI. | Operator clicks the "Execute Navigation Plan" button. |
| **Create Manipulation Waypoint** | Operator clicks the "Create Manipulation Waypoint" button in the wristwatch UI and a waypoint is placed in the scene. The operator can also reach into the virtual robot's gripper and press and hold the grip button on their controller to spawn and grab a new waypoint. Finally, an operator can create a waypoint by holding an existing waypoint and pressing the trigger button to clone it. | Operator clicks the "Create Manipulation Waypoint" button in the Manipulation panel and a waypoint is placed in the scene. |
| **Adjust Manipulation Waypoint** | Operator grabs a waypoint by reaching into a waypoint and pressing and holding the grip button on their controller. The operator can also drag the slider on a waypoint to set how opened or closed the robot's gripper should be at the waypoint. | Operator clicks a waypoint and uses marker controls (translational arrows or rotational scroll circle) to move the waypoint. |
| **Plan Manipulation Trajectory** | Operator clicks the "Plan Manipulation Trajectory" button in the wristwatch UI. | Operator clicks the "Plan Manipulation Trajectory" button. |
| **Execute Manipulation Plan** | Operator clicks the "Execute Manipulation Plan" button in the wristwatch UI. | Operator clicks the "Execute Manipulation Plan" button. |
| **Move Robot's Head** | Operator reaches into the head of the virtual robot, presses and holds the grip button, and drags a ball to the location they would like the robot to look at. | Operator clicks on sliders to pan and tilt the head in the head control panel. |
| **Open / Close Gripper** | Operator can drag the slider on the virtual robot's gripper to open and close the gripper. | Operator can drag the gripper slider to open and close the gripper in the head control panel. |
| **Raise / Lower Torso** | Operator can drag the slider on the virtual robot's back to raise and lower the torso. | Operator can drag the torso slider to raise and lower the torso on the torso panel. |
| **View Robot Camera Feed** | Camera panel is located in front of virtual robot. | Camera panels is located on the top right of the screen. |
| **View Point Cloud Feed** | Displayed in world. | Displayed in world. |
| **View Planner Status** | Status messages on wristwatch interface. The color of waypoints also change to indicate planner status. | Status messages on the manipulation and navigation panels. |
| **Plan Visualizations** | Virtual robot loops through plan. The path generated from the navigation planner is also displayed on the floor. | Virtual robot loops through plan. The path generated from the navigation planner is also displayed on the floor. |
| **Operator's Viewpoint** | Operator physically looks / moves around with VR headset on. Operator can also teleport by pressing and pointing right trackpad button. | Operator clicks the viewpoint window and moves the mouse around to move the camera, similar to popular 3D modeling software such as autocad. |

TABLE I
DESCRIBES HOW INFORMATION AND PLANNING ACTIONS ARE PRESENTED IN BOTH THE VR AND 2D INTERFACES. THE FIRST COLUMN CONSISTS OF ALL INFORMATION OR PLANNING ACTIONS THAT MAKE UP THE INTERFACES. THE FOLLOWING TWO COLUMNS DESCRIBE HOW EACH INTERFACE SPECIFICALLY PRESENTS ALL FEATURES.

4a: The rate of improvement within participants will be greater in VR compared to the 2D condition. Where a higher rate of improvement corresponds to decreasing planning times over trials. Specifically we expect to see an interaction between the interface and trial conditions on performance such that participants' performance times will show greater decreases between trials in the VR condition than in the 2D condition.

4b: There will be fewer collisions with the environment in the VR condition compared to the 2D condition. We expect that there will be fewer collisions in the VR condition because

the user will be able to adjust their vantage and waypoint positions easier in VR than in 2D.

4c: Participants in the VR condition will, on average, have higher task scores than those in the 2D condition. Task scores are determined by the equation:

Task Score = Number of Subtasks Completed - Number of Collisions

As participants in VR will experience fewer collisions

with the environment (H4b), they are also likely to complete more subtasks resulting in fewer points removed and higher task scores compared to 2D.

## IV. STUDY DESIGN

To compare our 2D and VR interfaces, we plan to recruit a total of 52 participants for the study, as determined by a power analysis conducted using the software G*Power [11]. Each participant will command a Fetch mobile manipulator robot [12] to complete several tasks using one interface type, either 2D or 3D. Participants will be pseudo-randomly assigned to each interface type while maintaining a proportional gender balance across interface conditions. Participants will then complete both a manipulation (block stacking) and navigation (obstacle avoidance) task using their assigned interface. In Figure 3 you can see the arena used for these tasks. The order of completion of these tasks will be randomly assigned and counterbalanced across participants to help control for learning effects across participants. Finally, each participant will complete three trials of each task and participants will have 15 minutes to complete each trial or as much of the trial as as they can. This design represents a 2 (Interface type) x 2 (Task type) x 3 (Trial) mixed design study, with Interface type as a between-subjects factor and Task type and Trial as within-subjects factors.

### A. Experimental Tasks

*1) Manipulation: Block Stacking:* In the block stacking task, participants will command the robot to stack three large Jenga blocks in a tower. An example of the tower participants will be asked to replicate will be in the participant's view as they complete the task. Three blocks representing the base of the tower, which the participant will stack blocks on will be provided as seen in Figure 4. Task performance points are gained by lifting a block off of the table's surface and then for correctly stacking the block in the tower. Points are lost for every collision with the environment.

*2) Navigation: Obstacle Avoidance:* In the obstacle avoidance task, participants will navigate the robot to three different locations within a 3.05 m x 3.05 m arena seen in Figure 3. At each location is a goal which includes two Landolt C visual acuity charts which are mounted on a pole as seen in Figure 5. Across locations, Landolt C charts are mounted at a different height from one another. This requires the user to adjust the height of the Fetch robot in order to read both Landolt C's at each location. In the middle of the arena, there are three obstacles which the user will need to ensure that the robot avoids as it travels to each goal. These obstacles include a traffic cone, a wet floor sign, and a tool box as seen in Figure 3. Participants will be informed that there "will be obstacles blocking the robot's path," and the participant will be able to view all obstacles for each trial from inside the respective interface. Obstacle locations will move from trial to trial, where the locations of obstacles for each trial will be consistent across all participants.

At either 90, 135, or 180 seconds after each user starts planning to their next goal, the participant will be interrupted to complete a Situation Awareness measure. Each participant will encounter each condition once. To prevent users from looking around to find the answer to these questions, they will moved to a virtual room in VR, during the pause for answering, where they will not be able to see the robot or environment. For the 2D condition, the interface will be blanked out, also preventing the user from accessing information regarding the robot and its environment. Task performance points will be awarded by reaching each navigation goal and for correctly reading off the vision test. Performance points will be deducted for every collision with the obstacles and environment.

### B. Measures

*1) Biographical data measure:* First, participants will be asked to respond with their age and gender identity. Next, they will be asked to rate and describe their experience with robots, 3D modeling software, and virtual reality. Finally, participants will be asked to report if they wear corrective lenses, if they are currently wearing their corrective lenses, and if they need corrective lenses because they are near or far sighted.

*2) Perspective taking:* Participants will be asked to complete the Perspective Taking/Spatial Orientation measure by Hegarty and colleagues [13], [14]. This test is a trait-based measure of one's ability to imagine different perspectives from different locations in space. In the literature on human individual differences in spatial skills, visual perspective taking (VPT) is conceptualized as a visuospatial skill associated with spatial orientation skills. Hegarty and Waller [13] described that "VPT skill involves the ability to make egocentric spatial transformations in which one's egocentric reference frame changes with respect to the environment, but the relation between object-based and environmental frames of reference does not change [15] (p.176)." Because completing the manipulation and navigation tasks has a strong spatial component which at times may require changing spatial reference frames—for instance switching between robot-centric and an exocentric points of view—-we will measure individual differences in this spatial skill to be used as a potential covariate in data analyses.

The Perspective Taking test is composed of 12 items where each item presents respondents with a picture depicting an array of objects and an "arrow circle" with a question about the direction between some of the objects. Respondents are asked to imagine that they are standing at one object in the array (which is denoted in the center of the arrow circle), and facing another object depicted at the top of the circle. Respondents are then asked to draw an arrow from the center of the circle in the direction of a third object from the perspective described prior. Participants are given 5 minutes to complete the 12 items in the test. Each item is then scored by finding the absolute deviation in degrees between participants' response and the correct direction to the target (absolute directional error). As such, this measure is reverse scaled as more directional error is indicative of worse performance than less directional error.

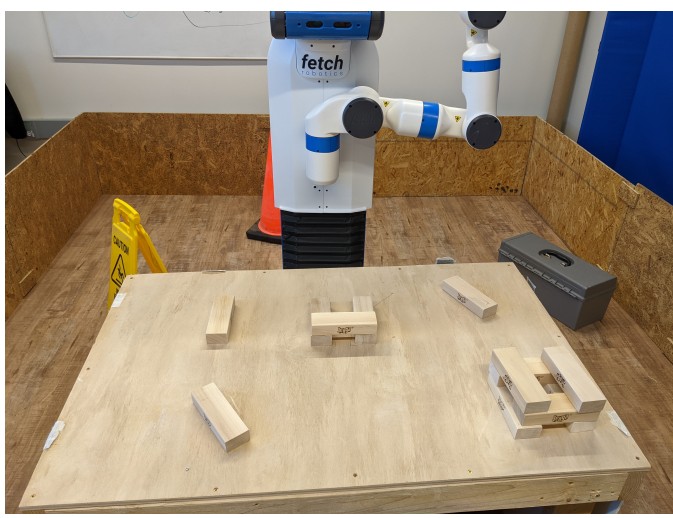

Fig. 3. The arena used in this user study. Here you can see the three obstacles for the navigation task including a traffic cone, wet floor sign, and toolbox. On the right side of the image you can see two of the navigation goals each with two buckets containing a Landolt C visual acuity chart. Lastly, in front of the robot you can see the table used for the manipulation block stacking task.

Fig. 4. A view of one table setup used for the manipulation block stacking task. On the bottom right is the example stack of blocks. In the middle, in front of the robot is the stack of blocks the participants will complete using the three single blocks on the table.

A participant's total score on this measure is given by the average deviation across all attempted items.

*3) Situation awareness:* Following the SAGAT method [16] for measuring Situation Awareness (SA), participant SA will be periodically probed while completing the navigation task. During each probe, participants will be moved into a virtual waiting room in the 3D condition or by blanking out the interface in the 2D condition. During this pause, participants will be asked to respond to SA questions that correspond to the three levels of SA: Level 1 (perception), Level 2 (comprehension), and Level 3 (projection, [17]). At each pause, one SA question from a bank of possible SA questions will be presented to participants and assigned randomly. Each question is designed to measure participants' situation awareness of the robot's state and its environment. Each question represents an objective measure of SA with only one correct answer. Participants will be paused and prompted to complete SA questions three times in each trial. The presentation of SA questions will be randomly timed beginning after participants start planning to a goal. The scoring of the responses to the SA questions will be as follows: correct responses to level 1 SA questions = 1

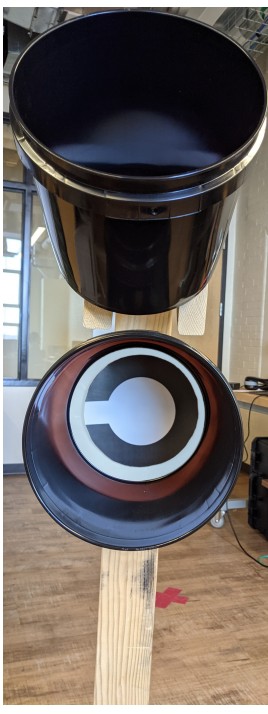

Fig. 5. A view of the lower Landolt C vision acuity test on a navigation goal.

point, correct responses to level 2 SA questions = 2 points, and correct responses to level 3 SA questions = 3 points. Points acquired for each SA question will be summed across trials of the navigation task and will be used to compare SA developed between 2D and 3D interfaces.

*4) Workload:* Two workload measures will be utilized: The NASA Task Load Index (NASA-TLX) and the Gas Tank Questionnaire.

The NASA-TLX [18] will be used to measure participants' perceived workload while completing tasks with their assigned interface. The NASA-TLX is a self-report measure for assessing workload associated with a variety of human-machine interfaces. Respondents provide ratings of their workload using six sub-scales: mental demand, physical demand, temporal demand, effort, frustration, and performance. The first five sub-scales are measured from 0 (Low) to 100 (High), and the sixth, performance sub-scale, is measured from 0 (Perfect) to 100 (Failure). The NASA-TLX also includes a weighted measure of paired comparisons among the sub-scales, but will not be included in this study because prior research has shown that workload scores obtained with and without the weighted sub-scales are often correlated above $r = 0.90$ [19]. Overall, workload scores will be calculated as the average of the six sub-scales after reverse coding scores on the performance sub-scale.

The Gas Tank Questionnaire (GTQ) [20] is a single item measure of remaining mental resources. Participants are asked to use the analogy of their brain as being like an engine and mental resources for completing tasks like gas/fuel for that engine. Participants are asked to think about "How much gas they have left right now" as a result of completing tasks. The GTQ was developed because prior research has suggested that repeated administrations of the NASA-TLX could result in "increasing workload simply by measuring it" [20]. To guard against confounding participant workload with repeated administrations of the NASA-TLX, we will measure workload using the GTQ after each task trial.

*5) Usability:* The System Usability Scale (SUS) [21] will be used to assess the perceived usability of each interface. The SUS includes 10 items covering different perceptions of the system like system complexity, consistency, and cumbersomeness. Participants rate items using 7-point Likert-type scales ranging from "strongly disagree" to "strongly agree." Each item on the SUS is converted and then combined, and scores are then multiplied by 2.5 to provide an overall usability score that can range from 0 (poor usability) to 100 (good usability).

*6) Performance:* To quantify performance while completing the manipulation and navigation tasks, we will record:

1) **Task Performance Points:** the participant's score on the sub-parts of the task. Points are gained for completing predefined subtasks, such as picking up a block or reaching the first navigation goal. Points are lost for each collision with the environment.
2) **Completion Time:** the overall time it takes participants to complete each task.
3) **Planning Time:** the amount of time a participant spends creating plans, which is determined from the time they create their first waypoint until the time they press execute.
4) **Number of Collisions:** the number of times the robot collides with its environment during a task.

*C. Procedure*

After filling out the Informed Consent form, participants will complete a training session for their assigned interface. This session walks participants through the interface and informs the user of each component of the interface as well as how to create, plan, and execute navigation and manipulation plans (a recording of the training session for the VR interface can be found below [1]). Once participants successfully complete the training session and are comfortable using their assigned interface, they will complete a baseline workload measurement using the NASA-TLX.

Participants will then complete three trials of the manipulation and navigation tasks according to their assigned interface condition and task order. After the first two trials of a given task (either manipulation or navigation), participants will complete the gas tank questionnaire. After the third trial of the task, participants will complete the SUS and another administration of the NASA-TLX. Then, participants will repeat this procedure for the three trials of their second task assigned via counterbalancing. After participants have completed all three trials of both the manipulation and navigation tasks,

[1]https://www.youtube.com/watch?v=bnNPh5dkTbE&list=PLfUzSIwyYwvUw0YTkqgNDsSsq-P8Ts8sV&index=1

they will complete the survey of demographic questions and complete the, perspective taking spatial orientation test. Once complete participants will be debriefed and compensated. All procedures will be reviewed by the University's Institutional Review Board.

## V. DISCUSSION

This study will be conducted in Spring 2022, if no further pandemic restrictions are implemented.

Through this study we will investigate the system usability, experienced workload, situation awareness, and task performance across manipulation and navigation tasks, as well as the skill acquisition rates across trials in our 2D and VR human-in-the-loop planning interfaces. This work will contribute to the literature by investigating VR interfaces for robot control beyond direct teleoperation.

By determining the aspects of each interface that contribute to better performance, our findings could potentially lead to the creation of hybrid (i.e., combinations of 2D and 3D) interfaces to improve human-in-the-loop control of robot systems. Ultimately, we hope to enable effective and intuitive teleoperation and supervision of robots to complete complex tasks in challenging environments.

## ACKNOWLEDGMENT

This work has been supported in part by the National Science Foundation (IIS-1944584 and IIS-1451427), the Office of Naval Research (N00014-18-1-2503), the Department of Energy (DE-EM0004482), and NASA (NNX16AC48A).

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
