# OpenReview forum: "Designing a User Study for Comparing 2D and VR Human-in-the-Loop Robot Planning Interfaces"
_humanrobotinteraction.org/HRI/2022/Workshop/VAM-HRI — VAM-HRI 2022_

### Official Review · Reviewer_RxQc · 2022-02-25
**Interesting and relevant paper, accept**

**Rating:** 9
**Confidence:** 5

**Review:**

The proposed user study to compare two human-in-the-loop planning interfaces (traditional 2D vs. 3D VR setup) is interesting and relevant to the VAM-HRI community. The paper is also well-written and motivated, and so this paper should be accepted. The reviewer has minor suggestions for the paper, but overall is excited to see the results of the proposed study.

Feedback:

1. It’s mentioned in Section II that the point cloud is visualized in Figure 1, but there does not appear to be any point clouds in Figure 1?
2. An image of the VR system would be highly beneficial to compare against Figure 2 of the traditional interface.
3. It is mentioned that obstacles will move from trial to trial. Will the way these objects be moved be consistent across users?

---

### Official Review · Reviewer_tUZi · 2022-02-26
**Thorough and relevant study proposed, accept**

**Rating:** 9
**Confidence:** 5

**Review:**

This paper describes a proposed study to compare a 2D interface using mouse and keyboard control with a VR interface using associated VR controls in a between-subjects study. Two tasks will be assigned to each participant, a manipulation task and a navigation task. There is strong motivation for this study with significant prior work to built upon. I recommend strong accept, and I provide some questions and other thoughts below that will hopefully assist the authors when carrying out the study.

- What prior work informed the interaction methods that you established (e.g. using buttons and sliders for the 2D interface)?
- In IV A 2, How is the "random moment" chosen for the SA assessment measure? What time span is provided?
- In IV B 1, Please use only "gender identity," not "preferred" (implying that gender is merely preference).
- These assessments are thorough, and many. You may want to consider how much time participants will be spending completing assessments vs. completing the tasks.

Minor typos:
Bottom of pg. 3 "Trail" should be "trial"
Section IV, B 4) Gas Tank Questionnaire should be abbreviated GTQ not GSQ (3 instances).

---

### Decision · Program_Chairs · 2022-03-04

Accept